# Non-Destructive Evaluation of Impacted CFRP by IR Thermography

**DOI:** 10.3390/ma12060956

**Published:** 2019-03-22

**Authors:** Waldemar Swiderski, Pawel Hlosta

**Affiliations:** Military Institute of Armament Technology, 05-220 Zielonka, Poland; hlostap@witu.mil.pl

**Keywords:** IR thermography, non-destructive testing, composite material

## Abstract

The aim of the article is to present a new technique providing an increase in the reliability of standard destructive tests of light ballistic shields. During the ballistic impact (i.e., of projectiles or fragments) on the material and its penetration by these incoming items, the absorbed kinetic energy is transformed into heat. In particular, the material regions that are damaged generate heat, and around and above the damage, on particular areas of the surface of the sample, the temperature signal increases. While registering, thermal cameras can process the impact and penetration of a material by a projectile and can accurately determine the area of the material (around the point of impact and the area of penetration) that has been damaged. Two infrared cameras were used for our testing work. One recorded the changes to the temperature field on the surface with the ballistic impact and the second one on the opposite surface. These results were compared with those obtained by optical active thermography performed by the reflection approach. Selected results from all the tests are presented in this paper.

## 1. Introduction

In military applications, laminates reinforced with aramid, carbon, and glass fibers are used for the construction of protective products against light ballistics. At the moment of impact into the laminate (in this case, a multi-layered carbon composite), a projectile is stopped (or not stopped) by a large number of individual fibers. As a result of the impact, fibers stretch and break to absorb the kinetic energy of the projectile casing. Moreover, the stretching of fibers in the fabric transfers the projectile’s energy to adjacent fibers and in this way, disperses the energy over a large area [1]. This creates a subsurface defect in the composite structure with a much greater area than the caliber of the projectile [2]. During the ballistic impact (i.e., of projectiles or fragments) on the material and its penetration by these incoming objects, the absorbed kinetic energy is transformed into heat. In particular, the material regions that are destroyed generate heat, and around and above the damage, on particular areas of the sample surface, the temperature increases.

The most effective methods of non-destructive testing (NDT) to detect damage in laminates are interferometry and infrared thermography [3]. NDT procedures using infrared thermography can be divided into passive and active methods [4,5]. In the passive method, the test object is characterized by the temperature field created during its ordinary functioning. In the active method, an external source of thermal stimulation (heating or cooling) is applied to the object. There are different techniques depending on the stimulation source, including pulsed, step, or modulated. The specimen is stimulated with an energy source, which can be of many types, such as heating lamps, lasers, eddy currents, microwaves, or ultrasounds [6]. Defects in test materials before the testing have a uniform temperature equal to ambient temperature, do not generate ‘useful’ temperature signals, and for this purpose require heating or cooling of the entire (or at least part of the) object. This testing creates a dynamic temperature field, and the results of its distribution depend on the observation time. IR cameras record the changes of the temperature field on the surface of the tested object. Two methods of observation are possible: reflection and transmission. For the reflection method, the thermal source and IR camera are located on the same side of the testing object. For the transmission method, the thermal source and IR camera are located on opposite sides of the testing object.

In this work, as a non-destructive testing method, step heating (long pulse) thermography was used to evaluate internal damage of the composite after destructive testing. Step heating has many applications, such as for coating thickness evaluation (including multilayered coatings), inspection of coating-substrate bonds, or evaluation of composite structures [4]. This method has been successfully used in the evaluation of composite material reinforced fibers [7].

There are known solutions using a thermal camera for recording areas of temperature increase at the point of impact and friction of the projectile in the target material [8,9]. They do not include events related to the assessment of internal damage to materials during their ballistic resistance testing. The new measuring system presented in this paper gives the possibility of choosing the non-damaged area of the sample of ballistic protection to be tested for further shooting. It is a completely new solution in ballistic resistance testing that is unheard of in the available literature.

## 2. Experimental Testing

### 2.1. Measuring System

The proposed solution [10] is a measurement system adapted to determine the area of internal damage in the composite material, which is made of a lightweight ballistic resistant cover, by non-destructive testing. In order to determine the ballistic resistance of the lightweight cover, a test was performed in which several shots were fired at a specimen of the cover. The commonly used evaluation is to determine the limit of V_50_ ballistic protection [10]. This is a determination of the arithmetic mean of the three highest values of projectile impact speed in the test sample resulting in partial penetration, and the three lowest values of impact velocity causing total penetration. This requires the shooting of several shots at the test sample. According to the requirements, the distance between successive projectiles is determined. As experimental research has shown using a new measuring system, the internal destruction area of material is much greater than the distance between successive projectiles that meet the requirements of the V_50_ method. Shooting a projectile into an area with internal damage falsifies the result of these tests. A projectile, on impact or full penetration of a test specimen, loses a significant part of its kinetic energy which is converted to heat and, in the penetration phase and shortly thereafter, accumulates in the material destruction area of the sample. This results in a change in temperature field on the surface of the sample. Similarly to Reference [11], infrared thermography was used in the monitoring of impact tests and the non-destructive evaluation of impacted specimens. Using an infrared camera in real time, we could accurately determine the destruction area of the sample material. This allowed the shooting of the next shot at a non-damaged area of the sample. When testing a sample made of low-conductivity materials with a thickness of more than a few millimeters (5–8 mm), it was preferable to use two thermal cameras. One camera recorded changes in the temperature field on the sample surface from the impact side of the projectile and the second camera on the opposite side. Figure 1 shows the experimental set-up using the new measuring system, which was equipped with two thermal cameras to detect temperature changes on both the impact surface of the projectile fired from the ballistic barrel and the opposite side of the sample. On the monitor, areas of temperature field changes were displayed on the surfaces of tested samples, and on this basis, the shooter could select an undamaged area of the sample for the next shot. The tests were carried out in a concrete shooting tunnel in which the ambient temperature was 17 °C.

### 2.2. Test Results

Tests were carried out on samples of composite plate made of five carbon fabric layers bonded with epoxy resin (UHU^®^PLUS 300, UHU GmbH&Co, Buhl, Germany). The carbon fiber plate (Carbon Center, prepreg compression molding, fiber orientation: 0°/90°, density: 1.58 g/cm^3^) had dimensions of 350 × 150 × 5.4 mm (Figure 2). Top Shot .22 LR ammunition was fired at the sample. A ballistic barrel was used to shoot the projectiles (2.15 g) with different velocities (calculated average kinetic energy of the projectile impact was about 83 joules for a velocity of about 278 m/s). At impact, changes in the temperature fields on both the surface impacted by the projectile and the rear surface were recorded by two thermal cameras. The thermal camera (FLIR SC 7600, FLIR Systems, Issy les Moulineaux Cedex, France) on the impacted surface was positioned such that the camera axis was perpendicular to the sample surface. The axis of the thermal camera (FLIR A650, FLIR Systems AB, Danderyd, Sweden) on the rear of the sample was directed at an angle of 60° to the sample surface so that the projectile could not damage the camera lens if it passed through the sample.

The parameters of the thermal cameras used in the tests are given in Table 1.

Figure 3 shows thermograms of a sample surface recorded during the V_50_ destructive test by the thermal camera on the impact side of the test specimen. For shot 1, the calculated average kinetic energy of the projectile impact was about 83 joules for a velocity of about 278 m/s. The areas on the sample surface at an elevated temperature indicate destruction to internal and external elements of the sample material. The thermograms (Figure 3) present the temperature field of the sample surface at 0.5 and 1 s after projectile impact.

Figure 4 shows the changes of temperature signal along the damaged subsurface area of the sample (line L) shown in Figure 3.

As shown in the graph, comparing obtained temperature waveforms with thermograms (Figure 3), the largest temperature increase on the surface of the sample occurring at the impact point of the projectile was over 30 °C (plot for 0.5 s). At this point, the decrease in temperature during the cooling of the sample (graph for time 1 s) was very small compared with changes in other areas where the temperature increase was greatest. In areas of greater damage (fiber breakage), the increase in temperature signal was from approximately 20 °C to 25 °C. The temperature on the right side of the main peak (pixel ~ 130) seemed to increase by almost 10 degrees between 0.5 and 1 s. This was due to a delayed temperature increase due to the thicker delamination in this place. A thicker layer of air filling the delamination caused this delay.

Figure 5 shows thermograms of the sample surface recorded from the exit side of the projectile. The thermograms recorded at 0.02, 1, and 5 s after impact of the projectile on the surface of the sample are shown. The thermogram reading taken at 0.02 shows very clearly visible places of detachment from the matrix of the composite bundles of carbon fibers. It is also optically visible. Graphs showing changes in temperature signal across the material destruction area are shown in Figure 6 (line L1 in Figure 5). At the point where the projectile pierced the sample, the temperature signal decreased. At the perforation point, there was a loss of part of the composite material, which resulted in a faster lowering of the temperature in this place. The increase in the temperature signal in the damaged area of the material after piercing by the projectile (0.02 s) was about 20 °C. In the initial cooling phase of the sample, there was a faster decrease in the value of the temperature signal (to 0.5 s). Later, this process was much slower.

Figure 7 shows sample thermograms from the impact side of the projectile shot at a speed about 4% higher than in the case shown in Figure 3. In this example, the area with the highest increase in temperature values (above 30 °C; Figure 7 and Figure 8) was larger than in the previous one (Figure 3 and Figure 4). A probable influence on the result could be the quality of joins between the layers of the composite made of epoxy resins. Moreover, when measuring the exit side of the projectile (Figure 9 and Figure 10), the characteristics of the course changes of the temperature signal was different. Line L1 was taken along the largest temperature rise on the thermograms (Figure 9).

## 3. Non-Destructive Testing

### 3.1. Experimental Set-Up

In order to verify and compare the results from the thermal cameras, after the ballistic testing, non-destructive testing of the samples was carried out via a step heating thermography method using a heat lamp with a power of 2 kW (heating time 3 s) and thermal camera (FLIR SC 7600). The total recording time was 10 s. A sequence of 1500 thermograms was recorded. The tests were performed using the reflection approach for the front (impacted) and rear (opposite) surface. The set-up used is shown in Figure 11.

### 3.2. Results

Compared with the pulsed IR thermography method, which is currently one of the most popular methods in non-destructive thermographic testing [12], in the step heating thermography method, the heating time of the test composite is much longer, while the power of heating energy is lower. The step heating method allows testing of materials with greater thickness than when using the pulsed method. Figure 12a shows the source thermogram made by step heating of the surface of the sample after destructive testing from the impact side of the projectile and illustrates the outline of an internal area of damage to the composite. Much better visibility of this area was achievable after phase analysis (Figure 12b) of this thermogram. Comparing the thermograms in Figure 3, Figure 7 and Figure 12, the sizes of the destruction areas of testing samples were very similar.

Pulsed phase thermography (PPT) has been successfully applied for defect detection purposes on a variety of materials. A great deal of work has been done to evaluate the potential of PPT for quantitative applications [13]. For this reason, this algorithm was chosen to analyze the sequence of thermograms. The Automation Technology IRNDT software was used for the PPT analysis.

A quantitative approach was used to compare the two methods (results from two thermal cameras and the step heating method). The purpose of the qualitative approach was to understand the phenomenon under investigation. The other goal used a quantitative approach that had better control. Therefore, to compare the two techniques, the quantitative approach was better.

In addition, the non-destructive test results shown in Figure 13 are consistent with the results shown in Figure 5 and Figure 6. The use of phase analysis (Figure 13b) of the source thermogram (Figure 13a) improved visualization of the internal damage area and perforation point of the sample by the projectile.

Non-destructive testing results from the step heating method test confirm the results obtained using the new measuring system in destructive test V_50_, as described in this paper.

The obtained results from both methods were very similar. For this reason, temperature profile analysis was not carried out over time. Temperature profiles over time showed changes in the temperature of a single point with very complex defects; a wrongly selected measuring point can only lead to incorrect conclusions.

## 4. Conclusions

When fired at a speed of 300 m/s, the kinetic energy of a projectile is about 50 J (shot with a standard fragment of 1.1 g), with some of this energy being converted into heat energy in the destruction area of the material. At the moment of impact into the laminate (i.e., the multi-layered carbon composite), a projectile is stopped by a large number of individual fibers. As a result of the impact, fibers stretch and break to absorb the kinetic energy of the projectile casing. This creates a subsurface defect in the composite structure with a much greater area than the caliber of the projectile [14]. This results in precise infrared imaging of damaged material areas. It is also possible to assess the degree of destruction of the internal material as more damaged areas generate more heat, and over these areas, the surficial temperature signal has a higher value. On the surface of the test sample only above the defect, a higher value of the temperature signal can even be observed for a few seconds (e.g., Figure 10). This has been demonstrated in numerous publications based on numerical calculations and experimental tests [15,16,17,18,19].

The analysis of composite materials after destructive tests using non-destructive IR thermography methods, despite the energy lamps used in these tests possibly being even greater than 6 kJ, does not obtain as accurate an imaging defect as in the proposed new method using two infrared cameras, since the heated lamp affects the entire surface of the test sample and cannot be used to generate precise heat only on the area of defect.

The tests using infrared cameras have shown that it is possible to determine the area of destruction of the internal structure of the composite carbon fiber reinforced plastic during destructive ballistic testing.

Future work will focus on the testing of composite structures with more layers and reinforced with glass and aramid fibers using ballistic testing in accordance with STANAG 2920 [20].

## Figures and Tables

**Figure 1 materials-12-00956-f001:**
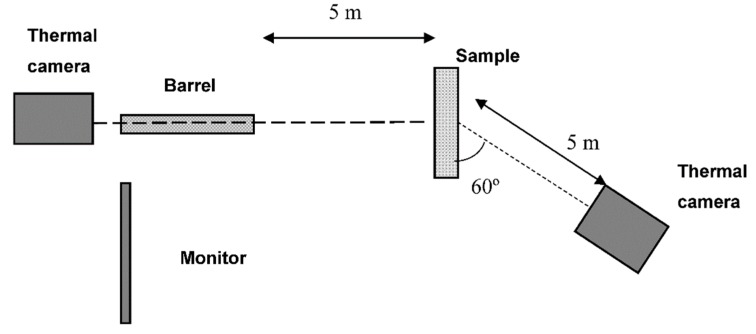
Experimental set-up.

**Figure 2 materials-12-00956-f002:**
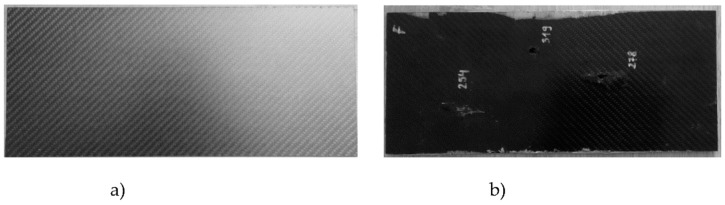
The sample (**a**) before destructive testing V_50_, and (**b**) after testing V_50_ (impact side).

**Figure 3 materials-12-00956-f003:**
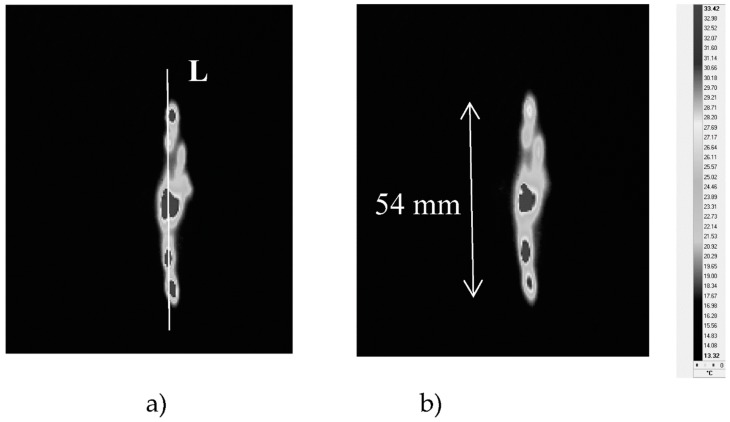
Thermograms of the sample surface after projectile impact: (**a**) 0.5 s, (**b**) 1 s.

**Figure 4 materials-12-00956-f004:**
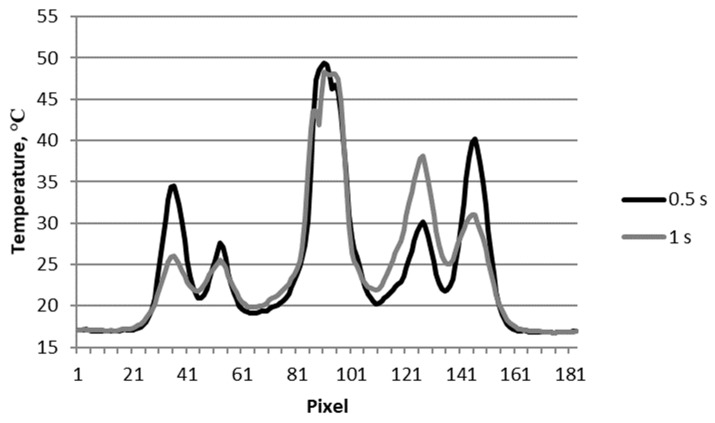
Temperature signal along the damaged subsurface area of the sample (line L) shown in Figure 3.

**Figure 5 materials-12-00956-f005:**
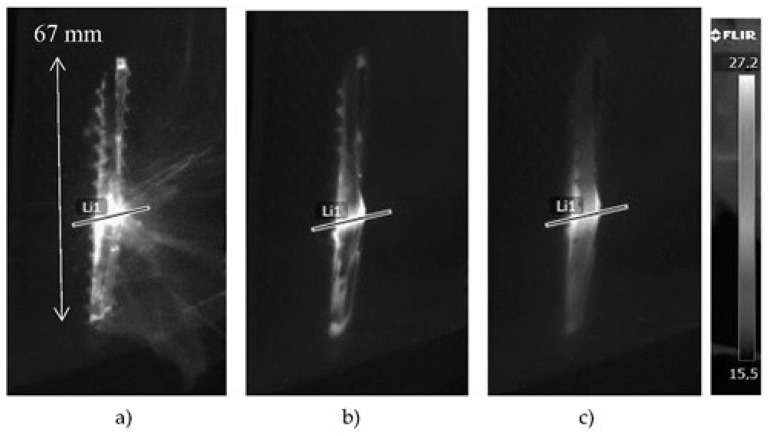
The thermograms recorded at (**a**) 0.02, (**b**) 1, and (**c**) 5 s after the impact of the projectile on the surface of the sample (the exit side of the projectile).

**Figure 6 materials-12-00956-f006:**
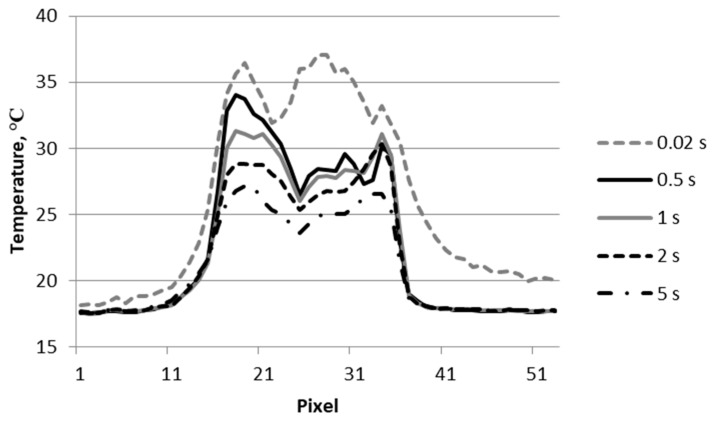
Temperature signal across the material destruction area (line L1 in Figure 5).

**Figure 7 materials-12-00956-f007:**
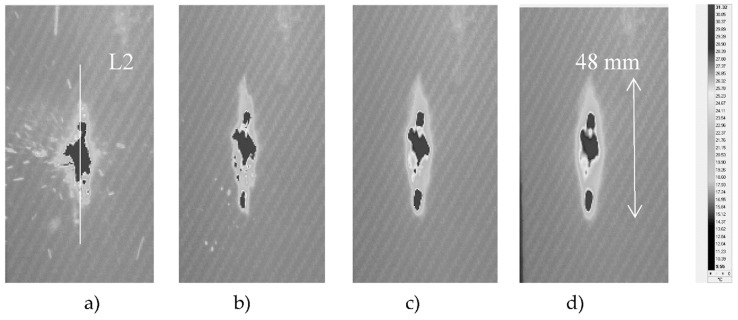
Thermograms of the sample surface after projectile impact: (**a**) 0.0 s, (**b**) 0.1 s, (**c**) 0.5 s, (**d**) 1 s.

**Figure 8 materials-12-00956-f008:**
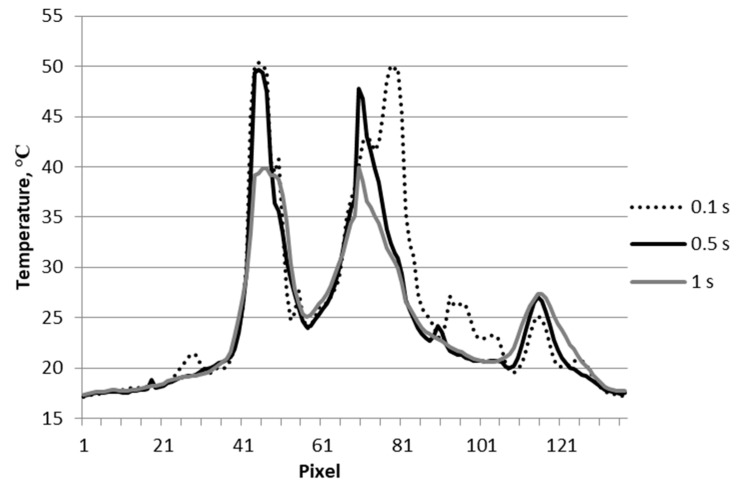
Temperature signal along the damaged subsurface area of the sample (line L2) shown in Figure 7.

**Figure 9 materials-12-00956-f009:**
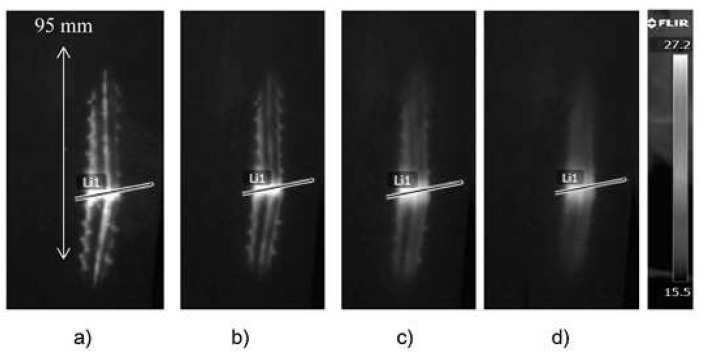
The thermograms recorded at (**a**) 0.02, (**b**) 0.5, (**c**) 2, and (**d**) 5 s after the impact of the projectile on the surface of the sample (the exit side of the projectile).

**Figure 10 materials-12-00956-f010:**
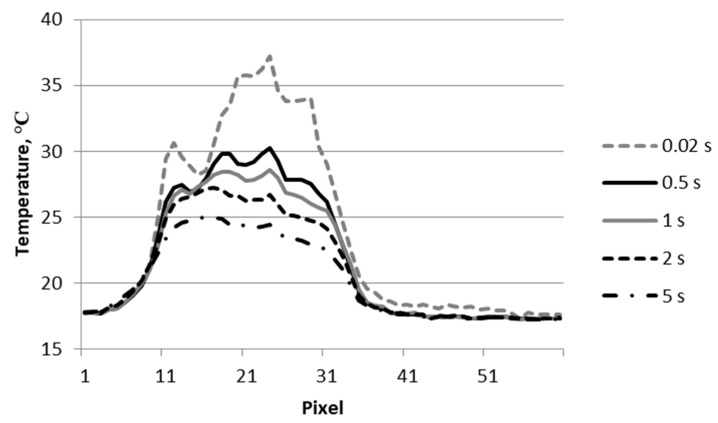
Temperature signal across the material destruction area (line L1 in Figure 9).

**Figure 11 materials-12-00956-f011:**
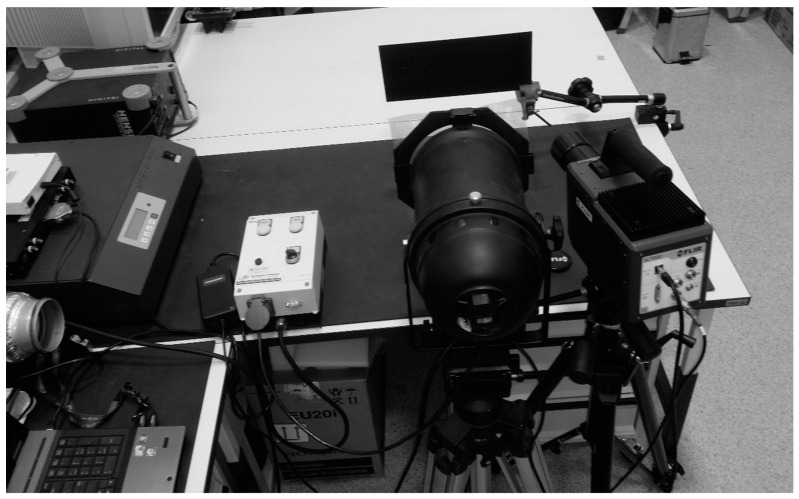
Experimental set-up.

**Figure 12 materials-12-00956-f012:**
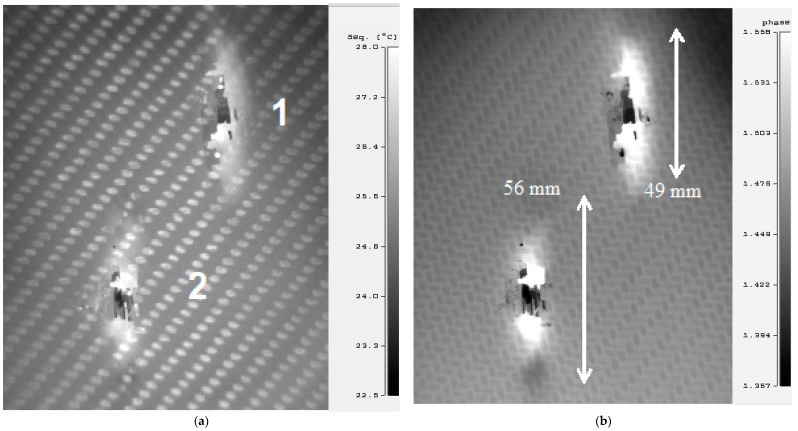
Thermograms made by step heating thermography of the impact side: (**a**) source thermogram, (**b**) after phase analysis.

**Figure 13 materials-12-00956-f013:**
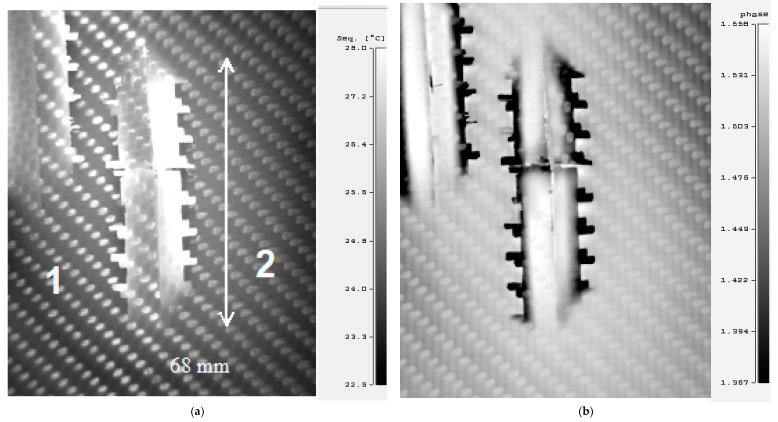
Thermograms made by step heating thermography of the exit side of the projectile: (**a**) source thermogram, (**b**) after phase analysis.

**Table 1 materials-12-00956-t001:** The parameters of thermal cameras.

Parameters	FLIR SC 7600	FLIR A655
Detector type	InSb	Uncooled microbolometer
Spectral range	3–5 µm	7.5–14.0 µm
Resolution	640 × 512	640 × 480
NETD (Noise Equivalent Temperature Difference)	17 mK	<30 mK
Detector pitch	15 µm	17 µm
Frame rate (full window)	380 Hz	50 Hz
Max frame rate (at min window)	30 kHz	200 Hz
Accuracy at reading	±1%	±2%

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
