# Peer review of "Non-Destructive Evaluation of Impacted CFRP by IR Thermography"

_materials, 2019, doi:10.3390/ma12060956_

Round 1

Reviewer 1 Report

The manuscript presents a new methodology to evaluate the internal damage generated by a ballistic impact on a CFRP laminate via IR Thermography. Knowing in real time the extension of the internal damage, guarantees that the all the impacts required for V50 evaluation are done on pristine areas of the sample. The work is systematic and very meaningful; however, it presents some minor issues that need to be addressed before publication.

1)     Please specify the type of carbon fibres and epoxy resin used for the manufacturing of the samples (e.g. manufacturer, weave pattern, gsm).

2)     Regarding the experimental set-up, it is not clear from Fig 1 how the camera can face the sample perpendicularly to the impact surface if the barrel is placed between them. Should it be then visible from the thermograms and obstruct the view?

3)     Please add a colour bar with the corresponding units in the thermograms of Fig 3, 5, 7, 9, 11 and 12. The lack of a scale makes difficult to understand exactly the extension of the internal damage, especially when comparing the results from Fig 3 and Fig 7.

4)     Please specify the velocity and energy associated with the two experiments. Section 2.2 indicates an energy of 83J for one of the impacts, while in the conclusions it is stated that the energy is about 50J.

5)     The thermogram in from Fig 4 clearly illustrates how the impact event generates heat within the sample which then cools down over time. However, the temperature on the right side of the main peak (pixel ~130) seems to increase by almost 10 degrees between 0.5 and 1s. Is this the effect of temperature building-up in the proximity of the damaged areas? Some considerations on this particular phenomenon should be included.

6)     Please use consistent line styles for the different curves of Fig 4, 6, 8 and 10. In particular, Fig 6 and 10 have the colour associated with the different times inverted, making the analysis of the data a bit confusing.

7)     The images from the step heating thermography (Fig 11 and Fig 12) should be adjusted as it is not clear which portion of the sample corresponds to the location of the first impact and which corresponds to the second. It is suggested to add labels on the images to indicate the location of the two different impact areas.  

8)     Fig 12 needs to be adjusted since only one damaged location is visible (is it the first impact location?) while the other is cut out from the picture. As a consequence, the damage observed in Fig 9 cannot be compared with the data recorded with the step heating thermography.

Typos:

Page 2, line 76. Please remove the full stop after “system” and join the two sentences. 

Author Response

Thank you for the comments:

1) Please specify the type of carbon fibres and epoxy resin used for the manufacturing of the samples (e.g. manufacturer, weave pattern, gsm).

Author response: Supplemented in the paper  - page 3 lines 92-93.

2)Regarding the experimental set-up, it is not clear from Fig 1 how the camera can face the sample perpendicularly to the impact surface if the barrel is placed between them. Should it be then visible from the thermograms and obstruct the view?

Author response: The camera lens is above the barrel. The angle between the axis of the barrel and the axis of the camera is only 4 degrees.

3)     Please add a colour bar with the corresponding units in the thermograms of Fig 3, 5, 7, 9, 11 and 12. The lack of a scale makes difficult to understand exactly the extension of the internal damage, especially when comparing the results from Fig 3 and Fig 7.

Author response: Supplemented in the paper.

4)     Please specify the velocity and energy associated with the two experiments. Section 2.2 indicates an energy of 83J for one of the impacts, while in the conclusions it is stated that the energy is about 50J.

Author response: Explained in the paper – page 3 line 95-97, page 4 line 123, page 6 lines 172-173, page 10 lines 239-240.

5)     The thermogram in from Fig 4 clearly illustrates how the impact event generates heat within the sample which then cools down over time. However, the temperature on the right side of the main peak (pixel ~130) seems to increase by almost 10 degrees between 0.5 and 1s. Is this the effect of temperature building-up in the proximity of the damaged areas? Some considerations on this particular phenomenon should be included.

Author response: Explained in the paper – page 5 lines 140-143

6)     Please use consistent line styles for the different curves of Fig 4, 6, 8 and 10. In particular, Fig 6 and 10 have the colour associated with the different times inverted, making the analysis of the data a bit confusing.

Author response: Supplemented in the paper.

7)     The images from the step heating thermography (Fig 11 and Fig 12) should be adjusted as it is not clear which portion of the sample corresponds to the location of the first impact and which corresponds to the second. It is suggested to add labels on the images to indicate the location of the two different impact areas.  

Author response: Supplemented in the paper.

8)     Fig 12 needs to be adjusted since only one damaged location is visible (is it the first impact location?) while the other is cut out from the picture. As a consequence, the damage observed in Fig 9 cannot be compared with the data recorded with the step heating thermography.

Author response: Supplemented in the paper.

Typos: Page 2, line 76. Please remove the full stop after “system” and join the two sentences.

Author response: Supplemented in the paper.

Reviewer 2 Report

In the paper a new non-destructive technique, based on IR thermography, is proposed for the assessment of the damage in CFRP composite materials subjected to impact loads. The subject of the paper is interesting and in the field of “Materials” journal.

General comment

The Authors should highlight the advantages in the use of the proposed technique and the novelty of the proposed technique with respect to those generally adopted in the literature.

Moreover, the following points should be considered:

Abstract

-        Page 1 line 17: replace “approache” with “approach”.

-        Please clearly highlight the objective of the paper and the novelty of the proposed technique with respect to those generally adopted in the literature.

1.      Introduction

-        Page 2 line 32: replace “temperature signal” with “temperature”.

-        Page 2 line 36: replace “a temperature” with “the temperature”.

-        The Authors should add a paragraph at the end of the Introduction Section to clearly state the objective of work and the experimental activities and the analysis reported in the paper. Moreover, the Authors should highlight the novelty of their work and the differences and the advantages of using the proposed techniques with respect to those adopted in the literature.

2.      Experimental testing

-        Page 2 line 55: remove the word “new”.

-        Page 2 line 56: replace “to the composite material” with “in the composite material”.

-        Page 2 line 57: replace “lightweight” with “the lightweight”.

-        Page 2 line 58-58: “The commonly used evaluation is to determine the limit of V50 ballistic protection” When is this technique used? Please add a reference to specify when this technique is used.

-        Page 2 lines 62-65: “Targeting a projectile into a material area with existing internal damage falsifies the result, defined as being the distance between successive projectile hits, however, as revealed by experimental studies using a new measuring system, the internal area of destruction of the material is much greater than the distance between successive adopted places of projectile impact.”. This sentence is not clear and should be rewritten.

-        Page 2 line 73: “with a thickness of more than a few millimetres”: What do the Authors mean with “few millimiters”? Please insert a numerical value.

-        Page 2 line 86: Please add a description of the “ballistic barrel” used or add a reference describing it. Add also the impact speed and the mass of the projectile.

-        Fig. 4 and Fig. 6: For the abscissa axis, it would be better to plot the distance in mm, rather than the number of pixel.

-        Page 5, line 141: replace”0,02” with “0.02”.

-        Page 5, lines 143-144: “Graphs showing changes in temperature signal (Figure 6) across the material destruction area (line 143 L1 - Fig.5) show the perforation point of the specimen by the projectile (decrease of temperature signal at this point)”. This sentence is not clear and should be rewritten. Why was the temperature decreasing in the perforation point?

-        Section 2 and Section 3 should be reorganized to improve its readability: An example could be:

o   Section “2.1 Materials”, in which the material properties are reported;

o   Section “2.2 Experimental configuration”, in which the testing setup is described.

o   Section “3.1 Experimental results” in which all the experimental results in Section 2 are reported.

o   Section “3.2 Non destructive testing”.

3.      Section 3

-        The Authors should expand this Section to confirm that the proposed technique permits to obtain the same results that can be attained through non-destructive techniques used in the literature.

-        Section 4, “Conclusions”: The Authors should avoid the use of references in the “Conclusions” Section.

Author Response

Thank you for the comments:

-        Page 1 line 17: replace “approache” with “approach”.

Author response: Corrected

-        Please clearly highlight the objective of the paper and the novelty of the proposed technique with respect to those generally adopted in the literature.

Author response: Corrected

-        Page 2 line 32: replace “temperature signal” with “temperature”.

Author response: Corrected

-        Page 2 line 36: replace “a temperature” with “the temperature”.

Author response: Corrected

-        The Authors should add a paragraph at the end of the Introduction Section to clearly state the objective of work and the experimental activities and the analysis reported in the paper. Moreover, the Authors should highlight the novelty of their work and the differences and the advantages of using the proposed techniques with respect to those adopted in the literature.

Author response: supplemented

-        Page 2 line 55: remove the word “new”.

Author response: Corrected

-        Page 2 line 56: replace “to the composite material” with “in the composite material”.

Author response: Corrected

-        Page 2 line 57: replace “lightweight” with “the lightweight”.

Author response: Corrected

-        Page 2 line 58-58: “The commonly used evaluation is to determine the limit of V50 ballistic protection” When is this technique used? Please add a reference to specify when this technique is used.

Author response: Corrected

-        Page 2 lines 62-65: “Targeting a projectile into a material area with existing internal damage falsifies the result, defined as being the distance between successive projectile hits, however, as revealed by experimental studies using a new measuring system, the internal area of destruction of the material is much greater than the distance between successive adopted places of projectile impact.”. This sentence is not clear and should be rewritten.

Author response: Corrected

– page 2 lines 71-75:

According to the requirements, the distance between successive projectiles is determined. As experimental research has shown using a new measuring system, the internal destruction area of material is much greater than the distance between successive projectiles that meet the requirements of the V50 method. Shooting a projectile into an area with internal damage falsifies the result of these tests.

-        Page 2 line 73: “with a thickness of more than a few millimetres”: What do the Authors mean with “few millimiters”? Please insert a numerical value.

Author response: Corrected – page 2 line 81

-        Page 2 line 86: Please add a description of the “ballistic barrel” used or add a reference describing it. Add also the impact speed and the mass of the projectile.

Author response: Corrected – page 2 lines 95-97

-        Fig. 4 and Fig. 6: For the abscissa axis, it would be better to plot the distance in mm, rather than the number of pixel.

Author response: Not changed

-        Page 5, line 141: replace”0,02” with “0.02”.

Author response: Corrected

-        Page 5, lines 143-144: “Graphs showing changes in temperature signal (Figure 6) across the material destruction area (line 143 L1 - Fig.5) show the perforation point of the specimen by the projectile (decrease of temperature signal at this point)”. This sentence is not clear and should be rewritten. Why was the temperature decreasing in the perforation point?

Author response: Corrected, page 5, lines 160 – 161:

Graphs showing changes in temperature signal (Figure 6) across the material destruction area (line 143 L1 - Fig.5). At the point of piercing the sample through the projectile, the temperature signal decreases.

-        Section 2 and Section 3 should be reorganized to improve its readability: An example could be:  Section “2.1 Materials”, in which the material properties are reported;  

  o   Section “2.2 Experimental configuration”, in which the testing setup is described. 

  o   Section “3.1 Experimental results” in which all the experimental results in Section 2 are reported. 

  o   Section “3.2 Non destructive testing”.

Author response: Corrected – section 3

-        The Authors should expand this Section to confirm that the proposed technique permits to obtain the same results that can be attained through non-destructive techniques used in the literature.

Author response: Corrected

-        Section 4, “Conclusions”: The Authors should avoid the use of references in the “Conclusions” Section.

Author response: Not corrected - discussion case

Reviewer 3 Report

The work regards the investigation of the ballistic impact on CFRP laminate by adopting two different approaches: passive and active thermography. In particular, the passive approach has been used for monitoring the heat energy during the impact while the active one for evaluating the damaged area.

The topic of the paper is interesting as well as the proposed approaches, but several changes are needed for improving the set-up description, the analysis and the results discussion.  

Detailed comments:

Firstly, aims are not clear in the abstract.

Introduction needs to be improved:

-Stepped approach has been used. Where is described this approach in introduction or in the paper ? Please refer to recent works and describe better within the paper as this technique works. Recent works:

   - Palumbo D, Galietti U, (2016), Damage Investigation in Composite Materials by Means of New Thermal Data Processing Procedures, Strain, 52(4), 276-285.

   - Palumbo D, Cavallo P, Galietti U, (2019), An investigation of the stepped thermography technique for defects evaluation in GFRP materials, NDT and E International, 102, 254-263.

Experimental set-up needs of more information (above all for the active approach):

-          Material information about the laminate lay-up,

-          Set-up of active approach (scheme or picture, test frequency,….),

Results:

-          Figures need the scale (temperature, phase, …). It is not clear what each figure represents (thermal contrast, phase,…).

-          How did you perform the signal processing ? Did you use the PPT algorithm ? Why ?

-          By using the step heating you can investigate both the heating and cooling. What did you investigate ?

-          Temperature profiles over time can provide more information. Why are not they in the paper ? Please, justify and discuss.

-          In order to compare the two techniques a quantitative approach is better than qualitative. Please, justify and discuss.

Author Response

Thank you for the remarks. I am sending the corrected article.

Firstly, aims are not clear in the abstract.

Author response: Corrected – page 1 lines 8-9

Introduction needs to be improved: -Stepped approach has been used. Where is described this approach in introduction or in the paper ? Please refer to recent works and describe better within the paper as this technique works. Recent works:  

- Palumbo D, Galietti U, (2016), Damage Investigation in Composite Materials by Means of New Thermal Data Processing Procedures, Strain, 52(4), 276-285.  

- Palumbo D, Cavallo P, Galietti U, (2019), An investigation of the stepped thermography technique for defects evaluation in GFRP materials, NDT and E International, 102, 254-263.

Author response: Corrected – page 2 lines 49-54, reference - 7

Experimental set-up needs of more information (above all for the active approach):

-          Material information about the laminate lay-up,

Author response: Corrected – page 3 lines 92-93

-          Set-up of active approach (scheme or picture, test frequency,….),

Author response: Corrected – Fig. 11, page 8 lines 192-193

Results:

-          Figures need the scale (temperature, phase, …). It is not clear what each figure represents (thermal contrast, phase,…).

Author response: Corrected

-          How did you perform the signal processing ? Did you use the PPT algorithm ? Why ?

Author response: Corrected – page 8 lines 212-216

-          By using the step heating you can investigate both the heating and cooling. What did you investigate ?

Author response: Corrected – lines 191-192

-          In order to compare the two techniques a quantitative approach is better than qualitative. Please, justify and discuss.

Author response: Corrected – lines 217 - 220

Round 2

Reviewer 2 Report

The paper has been revised according to my indications and all the points have been properly answered.

Author Response

Thank you very much for your comments. We have included them in our revised article.

-        Page 1 line 17: replace “approache” with “approach”.

Author response: Corrected

-        Please clearly highlight the objective of the paper and the novelty of the proposed technique with respect to those generally adopted in the literature.

Author response: Corrected

-        Page 2 line 32: replace “temperature signal” with “temperature”.

Author response: Corrected

-        Page 2 line 36: replace “a temperature” with “the temperature”.

Author response: Corrected

-        The Authors should add a paragraph at the end of the Introduction Section to clearly state the objective of work and the experimental activities and the analysis reported in the paper. Moreover, the Authors should highlight the novelty of their work and the differences and the advantages of using the proposed techniques with respect to those adopted in the literature.

Author response: supplemented

-        Page 2 line 55: remove the word “new”.

Author response: Corrected

-        Page 2 line 56: replace “to the composite material” with “in the composite material”.

Author response: Corrected

-        Page 2 line 57: replace “lightweight” with “the lightweight”.

Author response: Corrected

-        Page 2 line 58-58: “The commonly used evaluation is to determine the limit of V50 ballistic protection” When is this technique used? Please add a reference to specify when this technique is used.

Author response: Corrected

-        Page 2 lines 62-65: “Targeting a projectile into a material area with existing internal damage falsifies the result, defined as being the distance between successive projectile hits, however, as revealed by experimental studies using a new measuring system, the internal area of destruction of the material is much greater than the distance between successive adopted places of projectile impact.”. This sentence is not clear and should be rewritten.

Author response: Corrected – page 2 lines 71-75:

According to the requirements, the distance between successive projectiles is determined. As experimental research has shown using a new measuring system, the internal destruction area of material is much greater than the distance between successive projectiles that meet the requirements of the V50 method. Shooting a projectile into an area with internal damage falsifies the result of these tests.

-        Page 2 line 73: “with a thickness of more than a few millimetres”: What do the Authors mean with “few millimiters”? Please insert a numerical value.

Author response: Corrected – page 2 line 81

-        Page 2 line 86: Please add a description of the “ballistic barrel” used or add a reference describing it. Add also the impact speed and the mass of the projectile.

Author response: Corrected – page 2 lines 95-97

-        Fig. 4 and Fig. 6: For the abscissa axis, it would be better to plot the distance in mm, rather than the number of pixel.

Author response: Not changed

-        Page 5, line 141: replace”0,02” with “0.02”.

Author response: Corrected

-        Page 5, lines 143-144: “Graphs showing changes in temperature signal (Figure 6) across the material destruction area (line 143 L1 - Fig.5) show the perforation point of the specimen by the projectile (decrease of temperature signal at this point)”. This sentence is not clear and should be rewritten. Why was the temperature decreasing in the perforation point?

Author response: Corrected, page 5, lines 160 – 161:

Graphs showing changes in temperature signal (Figure 6) across the material destruction area (line 143 L1 - Fig.5). At the point of piercing the sample through the projectile, the temperature signal decreases.

-        Section 2 and Section 3 should be reorganized to improve its readability: An example could be:

o   Section “2.1 Materials”, in which the material properties are reported;

o   Section “2.2 Experimental configuration”, in which the testing setup is described.

o   Section “3.1 Experimental results” in which all the experimental results in Section 2 are reported.

o   Section “3.2 Non destructive testing”.

Author response: Corrected – section 3

-        The Authors should expand this Section to confirm that the proposed technique permits to obtain the same results that can be attained through non-destructive techniques used in the literature.

Author response: Corrected

-        Section 4, “Conclusions”: The Authors should avoid the use of references in the “Conclusions” Section.

Author response: Not corrected - discussion case

Reviewer 3 Report

Figures 12 and 13 need to be improved. The scale is not clear.

Please revise the completeness of references.

Author Response

Figures 12 and 13 need to be improved. The scale is not clear.

Please revise the completeness of references.

Author response: Thank you very much for your comments.

I improved the quality of the scale in Fig. 12 and 13. Position No. 7 in the literature is also added.